# Cost and outcome of behavioural activation versus cognitive behavioural therapy for depression (COBRA): a qualitative process evaluation

Katie Finning,[1] David A Richards,[1] Lucy Moore,[1] David Ekers,[2] Dean McMillan,[3] Paul A Farrand,[4,5] Heather A O'Mahen,[4] Edward R Watkins,[4] Kim A Wright,[4] Emily Fletcher,[1] Shelley Rhodes,[1] Rebecca Woodhouse,[6] Faye Wray[7]

► Prepublication history and additional material are available. To view these files please visit the journal online (http://dx.doi.org/10.1136/bmjopen-2016-014161)

[1]University of Exeter Medical School, Exeter, UK
[2]Tees Esk and Wear Valleys NHS Foundation Trust, Darlington, UK
[3]Department of Health Sciences, University of York, York, UK
[4]School of Psychology, University of Exeter, Exeter, UK
[5]Clinical Education Development and Research (CEDAR), University of Exeter, Exeter, UK
[6]University of York, York, UK
[7]University of Leeds, Leeds, UK

**Correspondence to**
Katie Finning;
k.finning@exeter.ac.uk

## ABSTRACT

**Objective** To explore participant views on acceptability, mechanisms of change and impact of behavioural activation (BA) delivered by junior mental health workers (MHWs) versus cognitive behavioural therapy (CBT) delivered by professional psychotherapists.

**Design** Semistructured qualitative interviews analysed using a framework approach.

**Participants** 36 participants with major depressive disorder purposively sampled from a randomised controlled trial of BA versus CBT (the COBRA trial).

**Setting** Primary care psychological therapies services in Devon, Durham and Leeds, UK.

**Results** Elements of therapy considered to be beneficial included its length and regularity, the opportunity to learn and not dwelling on the past. Homework was an important, although challenging aspect of treatment. Therapists were perceived as experts who played an important role in treatment. For some participants the most important element of therapy was having someone to talk to, but for others the specific factors associated with BA and CBT were crucial, with behavioural change considered important for participants in both treatments, and cognitive change unsurprisingly discussed more by those receiving CBT. Both therapies were considered to have a positive impact on symptoms of depression and other areas of life including feelings about themselves, self-care, work and relationships. Barriers to therapy included work, family life and emotional challenges. A subset (n=2) of BA participants commented that therapy felt too simple, and MHWs could be perceived as inexperienced. Many participants saw therapy as a learning experience, providing them with tools to take away, with work on relapse prevention essential.

**Conclusions** Despite barriers for some participants, BA and CBT were perceived to have many benefits, to have brought about cognitive and behavioural change and to produce improvements in many domains of participants' lives. To optimise the delivery of BA, inexperienced junior MHWs should be supported through good quality training and ongoing supervision.

**Trial registration number** ISRCTN27473954, 09/12/2011

### Strengths and limitations of this study

► This study explored participant experience of behavioural activation (BA) and cognitive behavioural therapy (CBT) for depression in the context of the largest trial to date of BA and CBT, enabling an in-depth qualitative analysis alongside effectiveness, cost-effectiveness and quantitative process analyses.

► This is the first qualitative study comparing participant experience of BA delivered by junior mental health workers with CBT delivered by professional psychotherapists.

► Although our purposive sampling method ensured that interviews were completed with a diverse selection of participants, the generalisability of our findings are limited to those who were eligible and agreed to participate in the COBRA trial, and participants who declined to take part in the qualitative study may also have had different views to those who agreed to be interviewed.

► Despite aiming to interview participants as soon as possible after completion of therapy, there were sometimes barriers to achieving this, with a small number of participants commenting that they found it difficult to remember specific aspects of their treatment by the time of their qualitative interview.

## BACKGROUND

Depression is a common and debilitating condition that is the second largest cause of global disability.[1] Lifetime prevalence is estimated to be 16.2%, and 53% of people who experience one episode either do not recover at all or have at least one further recurrence.[2 3] Globally, the effect of depression on total economic output is predicted to be US$5.36 trillion between 2011 and 2030.[4] Cognitive behavioural therapy (CBT) is a psychological therapy that is as effective as antidepressants[5] and recommended by the UK's National Institute for Health and Care

Excellence (NICE).[6] However, it is complex, its effectiveness is dependent on the ability of the individual therapist[7] and the costs of training and employing therapists limits access. There is, therefore, a need to find alternative treatments that are effective, easily implemented and cost-effective. Behavioural activation (BA) is a simple psychological treatment that aims to re-engage patients with positively reinforcing experiences and reduce avoidance behaviours.[8] Furthermore, in 2001 it was proposed that BA should be easier than CBT for patients and practitioners to understand and implement.[8] Although this idea remains to be directly tested, BA is an excellent potential candidate for further investigation and possible dissemination if it is shown to be equivalently effective compared with CBT.

COBRA is a randomised, controlled, non-inferiority trial of CBT delivered by professional psychotherapists compared with BA delivered by junior mental health workers (MHWs). Four hundred and forty patients with major depressive disorder were recruited through primary care and psychological therapies services in Devon, Durham and Leeds and randomised to receive up to 20 one-to-one, face-to-face sessions of BA or CBT delivered over 16 weeks, with the option of four additional booster sessions. The trial established that BA is clinically non-inferior to (ie, not worse than) CBT in reducing depressive symptoms and is more cost-effective.[9] Guidance from the UK Medical Research Council states that for effective interventions to be implemented it is necessary to understand not only if, but why the intervention works (ie, mechanisms of change) and how it can be optimised.[10] In addition, it is important to understand how acceptable treatments are to patients and what factors might influence adherence. Qualitative methods are recommended for the in-depth exploration of these issues and allow for examination of complex or unanticipated mechanisms and consequences of treatment.[11]

There have been few qualitative studies investigating patient experience of face-to-face CBT for depression and, to the best of our knowledge, none of BA. Barnes *et al*[12] found that patients receiving CBT for depression frequently struggled with homework, that aspects of the therapeutic relationship could be difficult and some patients reported that CBT did not adequately address the cause of their depression. However, despite these potential barriers patients reported finding CBT beneficial overall. Other research[13–15] has shown that patients receiving CBT for depression value both common factors (ie, elements of treatment that are shared across most interventions, such as the therapeutic relationship) and specific factors (ie, the particular theoretical orientation adopted by the therapist and the techniques based on that theory),[16] a finding replicated in studies of CBT for other conditions including anxiety,[17] eating disorders,[18] and psychosis.[19] Common factors that patients appear to particularly value include collaboration with the therapist and the opportunity to learn about their condition,[13–15 17–19] and important specific factors include

identifying and changing negative cognitions.[13 17] In addition, a recent study of individuals' long-term use of CBT skills found that patients who view CBT as a learning experience are more likely to report cognitive and behavioural change and to continue using the skills learnt long-term, compared with those who solely consider CBT an opportunity to talk about their depression.[20] The current study aimed to explore participant experience of BA and CBT by addressing the following research questions in a selection of participants in the COBRA trial:

1. What are participants' views about the acceptability of BA and CBT?
2. What are participants' views about the role of cognitive and behavioural change strategies?
3. What are the broader impacts of BA and CBT on participants' lives?

## METHODS

### Sample

At baseline assessment all COBRA trial participants were asked whether they would be willing to complete an additional interview at a later date to discuss their experiences of therapy. Participants for the qualitative study were selected from those providing consent and were purposively sampled to ensure a selection of participants from each recruitment site, both trial arms, some who had fewer than eight sessions of therapy and some who had eight or more. Of those having eight or more sessions, we purposively sampled some who remained depressed and others no longer depressed according to the Structured Clinical Interview for Diagnostic and Statistical Manual Version IV (SCID)[21] at 6-month follow-up. The sample size (n=36) was predetermined based on the selection criteria, to ensure recruitment of participants from each of our purposive sampling criteria described above. Participants were invited to take part in the qualitative study via letter, followed up by a phone call from a researcher. Interviews were conducted as soon as possible after therapy had ended. The main trial and nested qualitative study received Multi-Centre Research Ethics Committee approval from the South West Research Ethics Committee in the UK (ref 12/SW/0029). Full details of the main trial are reported in Richards *et al*.[8]

### Design

Semistructured interviews were conducted over the telephone by KF, RW and FW, who were researchers working on the COBRA trial in Devon (KF) and Leeds (RW and FW). All had psychology degrees and experience of mental health research and were given in-house training in qualitative interviewing, followed by assessment and feedback on their first interview from DR, trial chief investigator experienced in qualitative interviewing. Interviews were completed across-sites to ensure researcher blinding was maintained for main trial follow-ups, for example the researcher in Devon completed interviews with participants in Leeds. Participants therefore had no prior

**Table 1** Interview topic guide

| Topic of discussion | Probe areas |
| --- | --- |
| **General experiences of treatment**<br>You recently received a course of CBT/BA as part of the COBRA trial. Please tell me about your experiences of receiving treatment | - What it felt like receiving treatment<br>- Anything in particular they liked or found helpful<br>- Anything they didn't like or found less helpful |
| **Barriers to treatment**<br>We are interested in reasons why people might decide to attend some or all of their therapy sessions, including completing some exercises and maybe not others. Please could you tell me about your reasons for deciding to continue with or stop therapy? | - Personal contextual factors<br>- Specific therapy factors<br>- Therapeutic relationship factors<br>- Stages or exercises causing difficulty?<br>- Anything (else) that could have been done to overcome thesedifficulties |
| **Cognitive change strategies**<br>We are interested in your views on the role of therapy in changing your beliefs or the way you think, and any impact this may have had on your mood. Did the therapy have any effect on your beliefs or the way you think? | - Underlying beliefs<br>- Style of thinking<br>- Influence of the changes in the way they think on mood/depression |
| **Behavioural change strategies**<br>We are interested in your views on the role of therapy in changing your behaviour, and any impact this may have had on your mood. Did the therapy have any effect on your behaviour? | - Changes in specific behaviour, for example, avoidance,rumination<br>- Recognising triggers and changing behaviour in response tothem<br>- Influence of behavioural changes on mood/ depression |
| **Most important part of therapy**<br>What was the most important aspect of therapy for you? | - Therapeutic relationship<br>- Exercises/homework tasks |
| **Broader impact of treatment**<br>Please tell us about the impact the treatment had on you generally or in other aspects of your life. | - Thoughts and opinions on depression<br>- The way they feel about themselves<br>- The role of psychological therapies in the treatment of depression<br>- Impact of treatment on any other areas of life |

BA, behavioural activation; CBT, cognitive behavioural therapy.

knowledge of, or relationship with, their qualitative interviewer. A semistructured topic guide was developed by DR and KF based on the study aims and previous literature. Interview topics included general experiences of treatment, acceptability and barriers to therapy, cognitive and behavioural change and the impact of treatment. Probe questions were used where necessary to help participants elaborate on their responses. The topic guide was pilot tested on two participants and modified to include a question on important parts of therapy, and probe questions were refined. The full and final topic guide can be viewed in table 1. Interviews were audio-recorded with participants' prior consent and transcribed verbatim. Transcripts were not returned to participants for comment but were double-checked for accuracy by a second member of the research team. Field notes were made by the lead author after conducting or listening to each interview, and these contributed to the initial development of codes. Interviews were conducted separately from main trial follow-ups to avoid bias and encourage open communication.

## Analysis

Data were analysed by KF, LM and DR using a framework approach[22][23] with the assistance of NVivo 10 software (QSR International). Analysis began with familiarisation with the transcripts and the development of an initial coding framework, combining deductive themes from the topic guide and inductive themes emerging from the data. KF and LM coded three interviews independently to assess the reliability of coding[24] and meetings were held to discuss and refine emerging themes. Each fragment of text was coded and compared with other fragments under the same code to determine commonalities and differences between codes, in keeping with the constant comparison approach.[25][26] Data for each code was repeatedly reviewed to ensure conceptual consistency, until a final coding framework was produced that covered all relevant themes identified in the text.[26] Continually comparing each new fragment of data enabled the content and definition of codes to be refined. Alternative explanations or negative cases were identified, discussed and a consensus reached.[24] Data were then summarised in framework matrices, using participants' own language to maintain the integrity of original accounts. This allowed us to further search for comparisons within and between cases, and between groups (eg, CBT vs BA), as well as demonstrating which participants had not discussed certain topics (highlighted by blank 'cells' in the matrix) and allowing us to analyse the data in its entirety. In the final stage of analysis, KF and DR met to discuss the findings

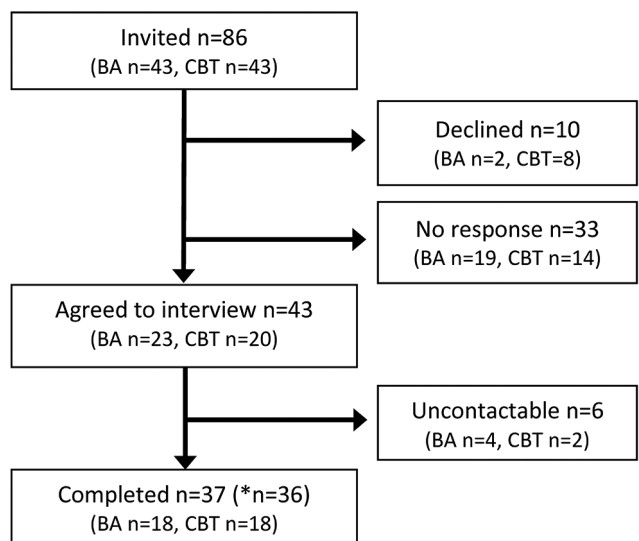

Figure 1 Flow of participants through the study. *Recording equipment failed for one interview, leaving 36 available for analysis. BA, behavioural activation; CBT, cognitive behavioural therapy.

in relation to the research aims and previous literature, focusing on drawing conclusions and synthesising the findings into the overarching themes presented in this paper. Analytical notes were kept throughout all stages of analysis to assist the interpretation of findings.

## RESULTS

Figure 1 shows the flow of participants through the study. Thirty-six interviews were completed between April 2014 and May 2015. Interviews lasted between 15 and 75 min and were completed on average 4 months after treatment ended (range 1–17 months). Participant demographics are provided in table 2, and the distribution of interviews across the purposive sampling frame can be seen in table 3.

Results are presented under three main headings: acceptability of therapy, mechanisms of change and impact of therapy, reflecting our three research questions. Table 4 provides examples of how analysis moved from the initial coding framework to the final themes and subthemes. Quotes are presented to support analysis and are labelled by participant ID number, therapy received, number of sessions, and for those who received eight or more sessions, depression status at 6-month follow-up (depressed or not depressed). Views were consistent across BA and CBT except where specified.

### Acceptability of therapy

Participants' views about the acceptability of therapy could be understood in terms of three subthemes: elements of therapy, the therapist and barriers to therapy.

### Elements of therapy

Many participants enjoyed therapy as an opportunity to learn about depression, themselves and their thoughts and behaviour. A few participants expressed a preference for one-to-one, face-to-face therapy over alternative modes, and for many the length and regularity of treatment was considered beneficial.

It's had a lasting effect and I think that may be to do with it being really quite in depth as you're going for an hour week… going once a week is helpful, which had been much better than going once every three months for six years, it's like you're really working on it, like a car. (28 - CBT 10 sessions, not depressed)

Some participants felt that therapy did not provide enough opportunity to talk about their feelings or the history behind their depression, but for others not having to focus on the past was considered helpful.

I really loved the fact that I didn't have to dwell on past experiences… I have seen therapists in the past and that but none of it's ever worked for me 'cos all they want to do is go over the past and I've never wanted to do that. (04 - BA 17 sessions, not depressed)

A small number (2/18) of participants in the BA group made additional comments that were not made by any of those receiving CBT. This included a resistance to the general BA approach, considering it simplistic, superficial and restrictive and that it was a 'poorer cousin' of CBT because of its lack of consideration to thought processes.

I feel like your life's more complicated or more complex than that… I think if you don't actually kind of go a little bit deeper and underneath things, you're just sort of tinkering around with some of the superficial stuff on the top and rearranging the furniture. (13 - BA 12 sessions, depressed)

In both treatments, experiences of homework were mixed. On the one hand it could be difficult, bringing therapy into everyday life and having the potential to make mood worse. Some felt a pressure to complete homework and it could create feelings of fear and failure. But for many, homework was considered an important part of therapy, providing them with a feeling of owning their depression and gaining control over their feelings, and having things written down was helpful. These views were not always distinct; some participants could recognise the benefits despite finding homework difficult.

People can give you information but you've got to put it into practice and act on it even if you might think 'Oh, this isn't going to really help me'… you've got to go through it and come out the other side, haven't you, to a certain degree? (07 - BA 13 sessions, depressed)

### The therapist

For many participants the therapist was a positive part of treatment: someone who was warm, patient and understanding. Participants in both treatments viewed their therapist as an expert who had the skills necessary to help them.

**Table 2** Demographics of qualitative participants

| Participant | Site | Therapy | No. of sessions attended | 6-month depression status | Gender | Age |
|---|---|---|---|---|---|---|
| 01 | Devon | BA | 1 | N/A* | Male | 63 |
| 02 | Devon | CBT | 3 | N/A | Female | 77 |
| 03 | Devon | BA | 14 | Not depressed | Male | 35 |
| 04 | Devon | BA | 17 | Not depressed | Female | 51 |
| 05 | Devon | CBT | 24 | Depressed | Female | 46 |
| 06 | Devon | BA | 3 | N/A | Female | 25 |
| 07 | Devon | BA | 13 | Depressed | Female | 52 |
| 08 | Devon | CBT | 22 | Depressed | Female | 36 |
| 09 | Devon | BA | 9 | Depressed | Male | 43 |
| 10 | Devon | CBT | 20 | Depressed | Female | 85 |
| 11 | Devon | CBT | 13 | Depressed | Female | 50 |
| 12 | Devon | CBT | 23 | Not depressed | Female | 62 |
| 13 | Devon | BA | 12 | Depressed | Male | 56 |
| 14 | Durham | CBT | 15 | Not depressed | Female | 22 |
| 15 | Durham | CBT | 14 | Not depressed | Female | 64 |
| 16 | Durham | BA | 13 | Not depressed | Male | 65 |
| 17 | Durham | BA | 2 | N/A | Male | 51 |
| 18 | Durham | CBT | 24 | Depressed | Female | 56 |
| 19 | Durham | CBT | 14 | Not depressed | Male | 44 |
| 20 | Durham | BA | 2 | N/A | Male | 49 |
| 21 | Durham | CBT | 19 | Depressed | Male | 58 |
| 22 | Durham | BA | 12 | Not depressed | Female | 50 |
| 23 | Durham | BA | 17 | Depressed | Male | 52 |
| 24 | Leeds | BA | 8 | Not depressed | Female | 35 |
| 25 | Leeds | CBT | 21 | Not depressed | Female | 55 |
| 26 | Leeds | BA | 24 | Depressed | Female | 37 |
| 27 | Leeds | CBT | 22 | Not depressed | Male | 40 |
| 28 | Leeds | CBT | 10 | Not depressed | Female | 58 |
| 29 | Leeds | BA | 24 | Not depressed | Male | 55 |
| 30 | Leeds | CBT | 3 | N/A | Female | 44 |
| 31 | Leeds | BA | 4 | N/A | Female | 39 |
| 32 | Leeds | BA | 5 | N/A | Male | 20 |
| 33 | Leeds | CBT | 1 | N/A | Male | 27 |
| 34 | Leeds | BA | 8 | Depressed | Female | 65 |
| 35 | Leeds | CBT | 10 | Depressed | Female | 39 |
| 36 | Leeds | CBT | 6 | N/A | Female | 25 |

*N/A: Depression status at 6-month follow-up was only included in the sampling frame for participants who had eight or more sessions of therapy.
BA, behavioural activation; CBT, cognitive behavioural therapy.

She was a lovely lady she gave me support when I needed it, she pushed me when I needed it… she could see when my mind was playing games with me where I was trying to ignore it or move around the situation, so for me she was very good. (04 - BA 17 sessions, not depressed)

For some, the therapist played a particularly important role in helping them overcome difficulties in therapy. Being able to adapt treatment, offering reassurance and not putting pressure on participants were helpful skills when therapy was difficult, and addressing challenges with the therapist was largely seen as a helpful process.

**Table 3** Number of interviews completed across the purposive sampling frame

| Therapy | Behavioural activation | | | Cognitive behavioural therapy | | |
|---|---|---|---|---|---|---|
| No of sessions attended | ≥8 | ≥8 | < 8 | ≥8 | ≥8 | < 8 |
| Depression status at 6 months | Depressed | Not depressed | N/A | Depressed | Not depressed | N/A |
| Devon | 3 | 2 | 2 | 4 | 1 | 1 |
| Durham | 1 | 2 | 2 | 2 | 3 | 0 |
| Leeds | 2 | 2 | 2 | 1 | 3 | 3 |
| Total | 6 | 6 | 6 | 7 | 7 | 4 |

Speaking to the counsellor and just being honest and open about what was, the fears or the barriers… because they were quite useful for her to understand, she could then fold that into the treatment as well. (27 - CBT 22 sessions, not depressed)

A small minority (2/18) of BA participants described their therapists as rigid, unauthoritative and lacking in confidence, comments that were not made by any of those receiving CBT. While only discussed by two participants in our sample, for them it appeared to be a significant problem and was discussed at length. One of these participants was also one of the two who was critical of the overall BA approach as discussed in the Elements of therapy section.

It did feel like there was a bit of a confidence issue going on or a lack of confidence from the therapist's side in some way. 'Cos I sort of picked up that I needed to kind of make her kind of feel like she was doing a good job with me. (13 - BA 12 sessions, depressed)

### Barriers to therapy

Work and family were particular features of life that could make therapy difficult. Getting to sessions could be problematic, particularly for those with comorbidities such as anxiety or chronic pain. A regular routine of appointments was considered to make attendance easier but flexibility was also welcomed, for example, rearranging sessions or completing them over the telephone if particular barriers arose. There were also emotional challenges to therapy; it could be hard to open up and talk about personal things, especially in the beginning and for those who were not used to expressing their emotions. For some, depression itself made it hard to put things into practice and affected their ability to understand components of therapy.

The whole point about mood which makes it bad is the fact that it's impacting your ability to do… the depression itself is a barrier to doing it. I actually can't offer a solution that would make it easier, but I believe that it wouldn't work for everybody. (33 - CBT 1 session)

Participants recognised the importance of their own attitude and commitment in helping them overcome barriers to therapy.

Hard as it was I was determined to do it because I knew I had to to make a difference in my life. (04 - BA 17 sessions, not depressed)

**Table 4** Worked examples of initial coding and the final themes and subthemes

| Interview extract | Initial codes | Final themes and subthemes |
|---|---|---|
| 'I felt the therapist was very good erm,she listened but she also asked questions and drew me out. I've um,I've done a counselling course in the past and had to see a counsellor who again,I was very pleased with,but she tended to just listen and I think, I think a lot of people need more input than just listening and what I liked about it is that they make very definite suggestions.' | Role of therapist Comparison to other therapies Important parts of therapy | Acceptability: The therapist Mechanisms of change: Talking vs doing |
| 'I mean the first thing was sort of um, being able to, um monitor and identify um, sources of stress an-and make me realise how, how I felt err how my behaviour changed and um the consequences of that. I've been able to identify the avoidance strategies that I'd put in place and then practically be able to um, deal with those problems straight away rather than um, um delay and avoid them. And by dealing with them in a shorter time span um they actually decreased um the anxiety that arose from them.' | Understanding and learning Triggers Avoidance Effect on symptoms | Acceptability: Elements of therapy Mechanisms of change: Behavioural change Impact of therapy: Impact for self |

## Mechanisms of change

Participants' views about mechanisms of change could be understood in terms of three subthemes: behavioural change, cognitive change and talking versus doing.

### Behavioural change

Changes to behaviour were considered important by many participants in both treatments, and this included avoidance, triggers, rumination and goal-oriented behaviour. Therapy-enabled participants to understand and overcome avoidance behaviours, and this could reduce anxiety.

> I felt like a weight had been lifted and I could, I was in a sort of procrastination phase where I couldn't make decisions and I was just puttin' things off… in gradual steps I started to be able to work my way through problems and being able to prioritise. (19 - CBT 14 sessions, not depressed)

Therapy helped participants recognise triggers for low mood and understand the consequences of their response to triggers. Some described being able to choose different behavioural responses, control their feelings with actions and think differently about triggers, and these changes could reduce the power of depression. Therapy helped participants learn to set realistic, achievable goals and use behaviour to improve their mood, as well as encouraging them to re-engage with positive activities and act when feeling low, helping to break the cycle of low mood.

> When people are depressed I know it's a circle, you don't do anything, so you feel terrible, and then you don't do anything… this therapy forced me, pushed me to act, to do something. This is the first time that I've actually experienced that somebody tell me 'Well let's start doing this and you will feel better' and it happened. (24 - BA 8 sessions, not depressed)

Both therapies encouraged participants to recognise the effect of rumination (ie, repetitive unproductive thinking, especially about the experience of depression) and for many, allowed them to manage and reduce time spent ruminating, which could improve mood and make life feel easier.

> I care about people, my family and friends, now. As I say I didn't before, I didn't want to go anywhere, didn't want to see anyone, I just wanted to be left alone. And that's the ruminating time; she got me off that and I feel better for it. (23 - BA 17 sessions, depressed)

### Cognitive change

Cognitive change was discussed by some participants in both therapies but was referred to more frequently by those who received CBT. This included having more self-belief, blaming themselves less when things go wrong and reduced beliefs of worthlessness.

> It's given me a different way of looking at things and I suppose that's the way of believing in things, I have more belief in myself, that has helped a lot. (12 - CBT 23 sessions, not depressed)

Participants in both therapies reported a more positive style of thinking, the ability to replace negative thoughts with positive and changing thoughts before entering a negative spiral. Other changes included a reduced tendency to overthink or ruminate, fewer self-critical thoughts and more balanced thinking. Some participants, particularly those who received CBT, described a sense of increased resilience such as the ability to reason when things go wrong and taking things less personally.

> I used to be a bit like a bull in a china shop if I was upset I would take it all very personally but now I'm more open minded… I think you don't take everything so personally, makes me think more rather than going to it head-long without thinking. (05 - CBT 24 sessions, depressed)

### Talking versus doing

This subtheme describes two typologies observed across both treatments, either prioritising opportunities to talk or using therapeutic strategies to bring about change. For several participants, having someone to talk to who was unbiased, non-judgemental and emotionally unconnected was the most important part of therapy, and problems from the week could be 'saved up' to discuss with the therapist.

> Irrespective of what therapy it was, I think just the opportunity for an hour a week to talk about how you're feeling was in some way therapeutic, irrespective of the specific techniques of the BA. (09 - BA 9 sessions, depressed)

In contrast, for many the 'doing' side of therapy was critical and the specific strategies of BA and CBT were considered helpful. The therapist was not just there to listen but to offer suggestions, and therapy was perceived to encourage participants to be proactive, finding a way to help themselves.

> I think a lot of people need more input than just listening and what I liked about it is that they make very definite suggestions… you'd look for evidence and look at what is happening and then make goals to try and work towards. (28 - CBT 10 sessions, not depressed)

## Impact of therapy

Participants' views about the impact of therapy could be understood in terms of three subthemes: impact for self, impact for others and impact on the future.

### Impact for self

In both therapies participants described no longer feeling depressed, enjoying life more and feeling like their old

self again, and for some these improvements were longer-lasting than they had experienced with other therapies.

> Now I don't feel so full of despair as I used to be. Sort of, oh it's like taming the beast, really… it's given me the tools to get through day-to-day life and be more aware of moods and what effect they have on me and how to change that mood. (12 - CBT 23 sessions, not depressed)

Other participants described themselves as happier as a result of therapy or feeling they have a different relationship with depression, leading to a feeling of acceptance. Several participants described therapy as having enhanced the way they feel about themselves, including increased feelings of self-compassion and improved self-esteem. Participants discussed positive influences of treatment leading to healthier lifestyles such as cooking better meals, exercising or seeking help for other problems like pain or disability. Treatment was believed by some to have enabled them to get jobs, return to work after a period of being signed off or perform better at work. For some, therapy enabled them to reduce or stop their antidepressants, and this could have further perceived benefits such as improved clarity of mind. Even those still meeting diagnostic criteria for depression could perceive a wide-reaching impact of therapy.

> I would just say thank you very much for all the help you've given me and it's made an impact on my life that I never would have thought. I thought I was on my own, but evidently I'm not. (23 - BA 17 sessions, depressed)

### Impact for others
Participants in both treatments discussed ways in which therapy influenced those around them. Many perceived therapy to have helped their relationships, and others described being more sociable or behaving more kindly towards others.

> We talk, which has never happened before. We talk for like, hours. And we don't need to watch the telly or listen to music or anything…so I'm more interested in what's going on than the one-eyed god. The television! (23 - BA 17 sessions, depressed)

### Impact on the future
Many participants described therapy as providing them with a 'toolkit' to take away, teaching them skills that have enabled them to deal with life more effectively. For some these skills were becoming automatic as they continue to put them into practice. Relapse prevention work was important, helping participants learn to recognise the signs of depression and knowing how and when to ask for help. Having paper copies of therapeutic tools to take away was considered helpful, and many participants revisit this when they feel low.

> I think that this will probably be something I'll do for the rest of my life, 'cos I'm sure that for the rest of my life I'll have the ups and downs like everyone else does, but this will stop me going back to those dark places. (11 - CBT 13 sessions, depressed)

## DISCUSSION
### Statement of principal findings
This study found that despite there being potential barriers to treatment, BA and CBT were perceived to have many benefits and were both considered to have brought about cognitive and behavioural change, leading, in participants' opinions, to improvements in specific symptoms as well as in their lives more broadly. Homework was considered an important part of treatment that allowed people to gain control of their feelings, but it could also be challenging. These views were not mutually exclusive; some participants could recognise the benefits of homework despite finding it difficult. This is consistent with previous research in which patients receiving CBT for depression reported finding homework difficult for both emotional and practical reasons, but that they understood its necessity in the therapeutic process.[12] The therapist played a significant role in helping overcome difficulties in therapy, and participants also recognised the importance of their own commitment and determination when therapy was hard. A small number (2/18) of BA participants commented that therapy could feel simplistic and that BA MHWs could appear rigid or lacking in confidence—views that were not expressed by any of those receiving CBT. This finding is of interest since our main trial results demonstrated that participants in each group remained in therapy for a comparable number of sessions (mean 11.5 BA sessions, 12.5 CBT sessions) and reported identical outcomes including recovery and remission rates.[9]

In both treatments there were two distinct views about mechanisms of change. For some people the most important part of therapy was having someone to talk to about their problems, and the therapist was someone they liked and regarded as an expert. This adds to the large body of research demonstrating that patients place high value on common factors such as the therapeutic relationship, and feeling understood and listened to by someone who is able to deal with their difficulties is considered important.[13 15 27] However, for many participants the specific therapeutic factors were fundamental. Given the theoretical orientation and aims of the two treatments, it is perhaps unsurprising that behavioural change was considered important for participants in both treatments, whereas cognitive change was largely discussed by those receiving CBT. This is, however, the first study providing evidence in BA for both common and specific factors and is consistent with previous qualitative studies of change processes in cognitive therapy, which have found that patients value both specific cognitive techniques such as changing negative thoughts and general psychotherapy

ingredients such as a collaborative therapeutic relationship and the opportunity to learn.[13–15]

BA and CBT both had potential for impact in broad and varied areas of participants' lives including improvements in mood, relationships, social contact, work and self-care. Therapy was considered to have provided people with skills for life and a toolkit for them to take away, and having a session on relapse prevention was helpful. Overall, despite finding therapy difficult at times, many participants found BA and CBT helpful in managing their symptoms of depression.

## Strengths and weaknesses

COBRA is the largest trial to date comparing CBT and BA and enabled an in-depth qualitative analysis of participants' experiences of these treatments alongside the effectiveness, cost-effectiveness and quantitative process analyses. This was also the first qualitative study comparing participants' experiences of BA delivered by junior MHWs with CBT delivered by professional psychotherapists. As the sample size was predetermined by our sampling criteria, no formal assessment of data saturation was performed. Therefore, we cannot exclude the possibility that richer accounts may have been obtained if more participants were interviewed. However, we believe that our purposive sampling method and the number of interviews completed minimised this possibility. Interviewing participants from a randomised controlled trial may limit the generalisability of our findings to the wider population since certain groups of patients may not meet eligibility criteria or self-select out of clinical trials like COBRA. Participants who declined to take part in the qualitative study may also have had different views to those who agreed to be interviewed. All interviews were completed over the telephone due to the long-distance nature of cross-site interviewing, thus maintaining researcher blindness for quantitative follow-ups, which was crucial for the integrity of the COBRA trial. Although some researchers have proposed that telephone interviewing may be less effective than face to face,[28] evidence suggests that telephone interviews yield the same amount and quality of data as those conducted face to face,[29] and some researchers argue that telephone interviewing may even be preferable when participants are discussing sensitive topics.[30]

Although we aimed to interview participants as soon as possible after completion of therapy, in practice this was difficult due to communication delays between therapists and researchers, difficulties recruiting participants quickly after therapy ended and an overall difficulty recruiting participants who had fewer than eight sessions of therapy, resulting in an extension of our recruitment period. A small number of participants commented that at the time of their qualitative interview they found it difficult to remember specific aspects of their treatment. It is possible that participants receiving fewer than eight sessions of therapy were less eager to be interviewed because they did not find therapy helpful and did not

wish to discuss those experiences with the research team. However, we successfully interviewed 10 participants in this category and are therefore confident that the views of those who dropped out of treatment were included in our analysis.

Finally, our analysis treated rumination as a habitual behaviour, consistent with the behavioural approach and the COBRA BA clinical protocol, which address rumination by activating the patient rather than using cognitive approaches to challenge the content of ruminations.[8] However, we recognise that rumination as repetitive negative thinking can also be conceptualised as a cognitive process, and our conclusion that cognitive change was largely discussed by participants receiving CBT could be interpreted differently depending on the theoretical approach taken.

## Implications

The COBRA trial results demonstrated the clinical non-inferiority and greater cost-effectiveness of BA delivered by junior MHWs compared with CBT undertaken by professional psychological therapists.[9] Data presented here from our qualitative process evaluation highlight the important role of therapists in helping patients overcome difficulties in therapy and suggest that therapists might be able to encourage people to remain engaged by offering reassurance, talking openly with them about difficulties experienced in therapy and adapting treatment to individual patients' needs. Participants also recognised the importance of their own commitment and determination when therapy was hard, and a conversation with patients about potential barriers to treatment may help to better prepare them for this. Many participants in both treatments described therapy as having provided them with skills for life, and relapse prevention was an element of therapy that was highly valued. This is consistent with previous research which has shown that even when patients make significant clinical improvements during therapy, they expect themselves to remain susceptible to depression and continue to implement techniques learnt as a way of managing what they consider to be a chronic condition.[31] This suggests that therapists should encourage patients to engage proactively in the therapeutic process and to consider it a learning experience, especially since previous qualitative studies have suggested that sustained improvements in depressive symptoms following CBT may be achieved by patients continuing to use the skills learnt in therapy.[20 31] The current study is the first to demonstrate similar evidence for BA and suggests that focusing on relapse prevention and providing patients with techniques to take away is a crucial component of both BA and CBT for depression.

Our findings also demonstrated that, despite BA being clinically non-inferior to CBT, BA could sometimes feel too simple, and MHWs could be perceived as rigid or lacking in confidence. Although in our study these comments were only made by those receiving BA, previous research has suggested that perceived inexperience of the

therapist can be a barrier for patients receiving CBT[12] and that dissatisfied CBT patients could consider their therapist to be delivering a rigid and predetermined therapy design.[32] It is possible that these criticisms were a function of BA itself, regardless of who was delivering treatment. However, since BA MHWs in the COBRA trial were less experienced than those providing CBT, we believe our results suggest that to optimise BA, junior MHWs require good quality training and ongoing supervision to encourage confidence in delivery of treatment, especially since inexperience and lack of confidence might impact negatively on the therapeutic relationship noted by our participants as critical to the therapeutic process.

### Unanswered questions and future research

This study has demonstrated that participant experience of the acceptability and impact of BA delivered by junior MHWs was similar to that of CBT delivered by professional psychotherapists, providing additional support to the argument that BA should be a front-line treatment for depression.[33] Our results also indicated that in both treatments patients value non-specific therapy factors such as the therapeutic relationship as well as tools and techniques specific to BA and CBT. While behavioural change was considered of high value by those in both treatments, cognitive change was discussed more frequently by participants who received CBT. This is unlikely to reflect a true difference in the mechanisms of change underlying the two treatments given that our parallel quantitative process evaluation did not detect such differences.[34] Rather, it is likely that participants were primed to talk about specific strategies by their experiences of therapy and the change strategies described by their therapist. We suggest that to further facilitate the delivery of BA by junior MHWs, their training should focus on the specific factors underpinning BA and on the importance of the therapeutic relationship, facilitating patients' opportunities to learn, preparing patients for homework, problem-solving difficulties in therapy and finally how to present a more confident therapeutic demeanour to provide patients with greater reassurance in worker expertise.

**Acknowledgements** We thank all participants, MHWs, therapists and general practitioners involved in this study, and Anthea Asprey for her contributions to transcribing. We acknowledge the contributions of study researchers and administrators in Devon, Durham and Leeds, the Peninsula Clinical Trials Unit and the NIHR Clinical Research Network.

**Contributors** DAR, DE, DMc, PAF, HAO'M, ERW and KAW designed the COBRA trial and were responsible for its conduct. SR, EF and KF were responsible for trial data collection management. KF, RW and FW collected qualitative data. KF, DAR and LM performed the qualitative data analysis. KF drafted the first version of the manuscript. All authors contributed and approved the final manuscript.

**Funding** This work was funded by the UK National Institute for Health Research (NIHR) Health Technology Assessment programme grant number 10/50/14. DAR is also supported by the NIHR Collaboration for Leadership in Applied Health Research and Care South West Peninsula.

**Disclaimer** The views expressed in this publication are those of the authors and not necessarily of the NIHR or UK Department of Health.

**Competing interests** All authors report grants from the UK National Institute for Health Research (NIHR) during the course of the study. DAR reports grants from the European Science Foundation. HAO'M reports grants from the UK Department of Health (DoH), Medical Research Council and Economic and Social Research Council and is an executive committee member of the BPS Perinatal Faculty and DoH Perinatal Clinical Reference Guideline. KAW is a clinical academic who teaches the theory and practice of BA and CBT.

**Ethics approval** South West Research Ethics Committee.

**Provenance and peer review** Not commissioned; externally peer reviewed.

**Data sharing statement** No additional data are available.

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
