## [Reviewer comments · BMJ Open]

ARTICLE DETAILS

TITLE (PROVISIONAL)	Cost and Outcome of Behavioural Activation versus Cognitive Behavioural Therapy for Depression (COBRA): A qualitative process evaluation
AUTHORS	Finning, Katie; Richards, David; Moore, Lucy; Ekers, David; McMillan, Dean; Farrand, Paul; O'Mahen, Heather; Watkins, Edward; Wright, Kim; Fletcher, Emily; Rhodes, Shelley; Woodhouse, Rebecca; Wray, Faye

VERSION 1 - REVIEW

REVIEWER	Sarah Hetrick The University of Melbourne Australia I have published a review of BA; I currently have a meta-regression under consideration where we have analysed the impact of BA only and cognitive approaches only on the overall treatment effect of CBT.
REVIEW RETURNED	24-Oct-2016

GENERAL COMMENTS	Overall, this is a beautifully written manuscript describing a well-executed add-on component to an important trial in the field. I have three main concerns with the paper. One concern was the reference to 'mechanisms' of change' in terms of the results that were described, particularly in the abstract. I don't think that there is enough emphasis or clear articulation of what the key change strategies are from the patient's perspective, and particularly with regard to what the implications are for clinicians in terms of what they should deliver (I think these were: understanding and overcoming avoidance – state of what; recognising triggers for low mood, thinking through consequences and choosing not to buy into the trigger; setting realistic goals; using behaviour to improve mood; recognising and managing emotions – but for all of these what would the clinician be delivering to achieve this? i.e. techniques). Making it very clear that these mechanisms of change are those things that the patient think are responsible for the improvements they experienced is important. I wonder whether COBRA measured potential mediators of outcomes as a way of considering what the mechanisms of change might be? The second concern was with the description of the methods with regard to the qualitative approach to analysis. This description is not as clear as it could be and there is virtually no description of the
--

constant comparative method e.g. the method according to Strauss and Corbin involves: 1. open coding where provisional codes are generated (and then I think that it is useful to give the reader some evidence of your method by describing what these were); 2. axial coding where examination of the shared characteristics of the descriptions with the same code results in categories being developed with criteria formulated for each category in order to facilitate comparisons between participant descriptions of their experience; 2. Selective coding: core categories are selected that best allowed for a description the nature of the data in terms of the categories that were started with to facilitate an explanation of what might determine acceptability, change strategies and impacts of CBT and BA. Some explanation of how this process makes it's a 'constant comparative method is required' e.g. the simultaneous process of coding and analyzing units of data by continually comparing each new unit of data facilitates a process of ongoing refinement of the content and definition of categories and how these relate to each other.

How this process relates to the Framework approach is also required.

As mentioned, I strongly believe it is important and useful for the reader to see more data with regard to the process of qualitative research so that either in the methods, or as part of the results the initial codes and frameworks that are developed and how these relate to the themes should be presented (often this is done in a table).

The third concern was with regard to whether the results are reflected appropriately in the conclusions. The opening sentence of the last section of the discussion (Unanswered question and future research) and the abstract states that the study has found BA to be acceptable delivered by junior MHW's but this brushes over the finding that in fact participants had some criticisms of this. I think the conclusion should focus on what the recommendations are i.e. need supervision etc....a focus on ensuring good engagement with a focus in training on the non specific aspects of therapy as critical, to include psycho education to prepare clients for the difficulties in therapy , and to engage them in a proactive therapy process. I would be keen to see more of the detail of what should be delivered by therapist as part of the implications, particularly given the stated focus on understanding the mechanisms of change or change strategies. The conclusions section of the abstract need to clearly link to the objectives of the study.

Minor points for revision

In the background there is reference to other similar qualitative research and I wonder if the authors came across anything about what techniques participants in previous research they perceived as being most useful?

There is reference in the background to patients who view CBT as a learning experience gaining more from therapy than those who use it as an opportunity to talk, and I wonder if the authors could reflect of their results with regard to these findings in the discussion (there is a result described as patients appreciating the opportunity to learn so the implication is that if they see this as a benefit that clinicians should maximise this; place emphasis on therapy being the

	opportunity to learn because it has been previously shown that this results in better outcomes than simply using therapy to talk). It is a little unclear how this finding relates to the rationale/objectives and findings of the current study. The authors refer to 'the challenges of therapy' a lot, but it isn't clear what they are referring to from the results – understanding challenges wasn't a key objective so the finding with regard to the objective they relate to (is this acceptability and is it related to the theme of barriers?). I think generally the point is to make sure that the key findings relate to the key objectives so you orient the reader and the reader can be sure that you have addressed the objectives. The implications section of the discussion needs to more directly talk about the implications for clinicians – I think this is where the paper has a lot to offer; the implications need to be the focus rather than interpreting the findings from this and other research e.g. what does it mean in terms of implications that patients expect themselves to remain vulnerable and continue to implement techniques learnt during therapy? I think it is important to spell this out for the reader. I think the phrase with regard to the cognitive techniques "was largely discussed by participants who received CBT" (it is stated in the discussion and abstract) needs to be revised; it is expected that only these participants would refer to cognitive techniques, perhaps add this by way of revision. In terms of the limitations with regard to interviewing as soon as possible; I wonder if the other point is that you had difficulty getting those who had completed less than 8 sessions because they dropped out and didn't want to be involved and you could speculate about why this was e.g. it wasn't helpful? References Strauss A, Corbin J. Basics of Qualitative Research: Grounded Theory Procedures and Techniques. Newbury Park, CA: Sage Publications; 1990
--	--

REVIEWER	David M Clark University of Oxford, UK
REVIEW RETURNED	11-Nov-2016

GENERAL COMMENTS	This study explores the subjective experience of patients who received a course of behavioural activation (BA) or cognitive-behavior therapy (CBT) for depression. It is now fairly well established that these two therapies, which are both from the broad cognitive behavioural school of psychotherapy, have roughly equivalent effects in terms of symptom reduction. However, as far as I am aware, there are no comparative studies of patients' experiences while going through these two therapies, which differ in emphasis, if not orientation . Patient experience is an important topic, which deserves a place in the psychiatric literature. For that reason, this study is of interest. The authors use an established qualitative methodology and apply it in a generally appropriate manner. The results are mainly reported in the way that one would expect. However, in its present form the
---

manuscript suffers from some omissions in the Background, somewhat partial reporting of some of the Results, and statements in the Discussion that go beyond the evidence. These issues could all be rectified in a revision.

Background

The authors cite five previous studies that have explored patients' experience of CBT for depression. There are also a fair number of studies that have looked at patients' experiences of CBT for anxiety disorders and other conditions. In order to give the reader a fairer impression of the state of the field, it would have been good if these studies were briefly mentioned. The conclusions are in general similar to the ones that pertain to the studies of CBT for depression.

As far as I am aware, there are no quantitative studies of patients' experience behavioural activation. This is a clear selling point for the present study.

Results

It seems that patients report broadly similar experiences as they progress through BA and CBT. However, there are two areas where differences emerge (see pages 9 and 10). The way these differences are described betray the principal investigator's well-known preference for behavioural activation. I would suggest that the authors consider somewhat different phrasing so that a more impartial account is presented. In qualitative methodology one does not usually give percentage figures for the prevalence of particular themes. Instead, the convention is to use just a few, rather vague, adjectives to denote frequency. For example, "all", "many/most", "some/few" and "none". In general the authors stick to this convention. However, in the two places where a somewhat negative theme emerges for behavioural activation that is not present in CBT, the authors use a different language that would appear to be designed to minimise this observation. For example, "a small number" and "a small minority". This terminology is not used elsewhere in the Results section. I suggest the authors revert to using the more general terms that they use for other results. Alternatively, if they feel strongly that they want to quantify these particular observations, they should give the percentage figures. That would not be the normal convention for qualitative work and would seem special pleading. However, if the authors are particularly keen for readers to know the absolute rates that would be a reasonable alternative way of progressing.

One of the questions in the qualitative interview enquires about behaviour change and another enquires about cognitive change. The Abstract and the Results claim that BA and CBT are both perceived as promoting behaviour change but only CBT promotes cognitive change. This observation seems to rely on a particularly restrictive definition of cognitive change, which is not made explicit in the manuscript but is fairly common among researchers who have a strong allegiance to behavioural activation. The issue is how one categorizes rumination. The authors categorize it as a behaviour, which is how it is usually conceptualised within behavioural analysis. However, most cognitive psychologists and cognitive therapists would classify it as a cognitive process. It would be helpful if this well-known difference in conceptualisation were acknowledged.

	Background and Discussion The authors mention (page 3) their well-known belief that behavioural activation is easier than CBT to teach to relatively inexperienced and low paid mental health workers. This seems a rather pessimistic view of the intellectual abilities of a very important workforce and, as far as I'm aware, it has never been tested. Given this point, the statement should either be removed or qualified by indicating that it is a belief of the authors that has not been tested. The patients who formed the sample in this study are recruited from the COBRA trial, which has a confounded design. The trial seems to have been designed to demonstrate that behavioural activation is more cost-effective than CBT. As the two treatments have frequently been shown to achieve similar clinical outcomes, the design more or less guaranteed the cost-effectiveness result that the investigators were looking for by using high paid clinicians to deliver CBT and lower paid clinicians to deliver behavioural activation. A scientific test of the cost effectiveness issue, rather than a demonstration of a particular point of view, would have required an additional cell in which low paid workers delivered CBT. For the current paper, the problem with the confounded design is that it makes it difficult to interpret the occasional differences in findings between BA and CBT. Are the few negative comments relating to BA but not CBT a true reflection of the experience of BA or a function of the therapy being delivered by a relatively less experienced workforce? The authors seem to hold the view that therapist experience and training is the key issue. That might be correct. However, the contrary view should also be acknowledged as a possibility, especially as there are no other qualitative analyses of behavioural activation. The authors state in the Discussion that the perceived focus on behavioural and cognitive change in the treatments led to the observed symptom reductions. This statement goes beyond the data in the study. In order to demonstrate that cognitive or behavioural change mediates symptom change, a different quantitative methodology is required, with multiple measurement of the mediators and statistical modelling that relates the mediators to symptom outcome with suitable temporal lags. There are a fair number of studies that have done exactly this in CBT for depression and there may now also be some such studies for behavioural activation. If the authors want to link behaviour or cognitive change to symptom change, they should refer to such studies. Alternatively, if their statement is simply about patient's perceptions of the effectiveness of behavioural or cognitive change, they should make this clear. Further evidence for these perceived relationships should also be included in the results.
--	--

VERSION 1 – AUTHOR RESPONSE

Reviewer comment: One concern was the reference to 'mechanisms' of change' in terms of the results that were described, particularly in the abstract. I don't think that there is enough emphasis or clear articulation of what the key change strategies are from the patient's perspective, and particularly with regard to what the implications are for clinicians in terms of what they should deliver (I think these were: understanding and overcoming avoidance – state of what; recognising triggers for low mood, thinking through consequences and choosing not to buy into the trigger; setting realistic goals; using

behaviour to improve mood; recognising and managing emotions – but for all of these what would the clinician be delivering to achieve this? i.e. techniques). Making it very clear that these mechanisms of change are those things that the patient think are responsible for the improvements they experienced is important. I wonder whether COBRA measured potential mediators of outcomes as a way of considering what the mechanisms of change might be?

Authors' response: We have amended the abstract and discussion to make it clearer that the mechanisms of change we refer to are the patients' perspectives of what led to change in symptoms (p2,16). We have also amended the discussion to make it clearer how our results relate to implications for clinicians (p19, 20). The COBRA trial did include a quantitative mediation analysis, the results of which will be published elsewhere but were not within the scope of the current paper. We have added a statement relating to this in the discussion (p20).

Reviewer comment: The second concern was with the description of the methods with regard to the qualitative approach to analysis. This description is not as clear as it could be and there is virtually no description of the constant comparative method e.g. the method according to Strauss and Corbin involves: 1. open coding where provisional codes are generated (and then I think that it is useful to give the reader some evidence of your method by describing what these were); 2. axial coding where examination of the shared characteristics of the descriptions with the same code results in categories being developed with criteria formulated for each category in order to facilitate comparisons between participant descriptions of their experience; 3. Selective coding: core categories are selected that best allowed for a description the nature of the data in terms of the categories that were started with to facilitate an explanation of what might determine acceptability, change strategies and impacts of CBT and BA. Some explanation of how this process makes it's a 'constant comparative method is required' e.g. the simultaneous process of coding and analyzing units of data by continually comparing each new unit of data facilitates a process of ongoing refinement of the content and definition of categories and how these relate to each other.

How this process relates to the Framework approach is also required. As mentioned, I strongly believe it is important and useful for the reader to see more data with regard to the process of qualitative research so that either in the methods, or as part of the results the initial codes and frameworks that are developed and how these relate to the themes should be presented (often this is done in a table).

Authors' response: We have amended the methods to provide further description of the constant comparative method as described by Boeije (2002), and how this relates to the framework approach (p7). We have cited the reference by Boeije as opposed to the suggested Strauss and Corbin, as the latter is more strongly associated with Grounded Theory; a methodology that we did not use in the current study. We have added a table in the results section to demonstrate examples of initial codes and how they mapped onto final themes and subthemes (p9).

Reviewer comment: The third concern was with regard to whether the results are reflected appropriately in the conclusions. The opening sentence of the last section of the discussion (Unanswered question and future research) and the abstract states that the study has found BA to be acceptable delivered by junior MHW's but this brushes over the finding that in fact participants had some criticisms of this. I think the conclusion should focus on what the recommendations are i.e. need supervision etc....a focus on ensuring good engagement with a focus in training on the non specific aspects of therapy as critical, to include psycho education to prepare clients for the difficulties in therapy , and to engage them in a proactive therapy process. I would be keen to see more of the detail of what should be delivered by therapist as part of the implications, particularly given the stated focus on understanding the mechanisms of change or change strategies. The conclusions section of the abstract need to clearly link to the objectives of the study.

Authors' response: In response to comments from both reviewer one and reviewer two, we have amended the statements regarding criticisms of BA to highlight the fact that these were minority views expressed by only two out of the 18 BA participants interviewed. We therefore believe our conclusions

regarding the overall acceptability of BA and CBT to be a valid conclusion that does relate directly to the objectives of the study. We have, however, removed the more speculative comments about mechanisms of change in the 'unanswered questions and future research' section (p20), and have referred to the findings of our quantitative process analysis which will be included in our full trial report, which has been submitted for publication. We have also amended the unanswered questions and future research section to focus more on implications for clinicians, as requested.

Reviewer comment: In the background there is reference to other similar qualitative research and I wonder if the authors came across anything about what techniques participants in previous research they perceived as being most useful?

Authors' response: The studies referenced in the background did not discuss specific therapeutic techniques, but did discuss general elements of therapy that participants found helpful, namely collaboration with the therapist, learning (about themselves, about depression, about thoughts), and being able to change negative thoughts. We have amended the introduction to include a description of these findings (p4), and have also referred to them in the discussion (p17).

Reviewer comment: There is reference in the background to patients who view CBT as a learning experience gaining more from therapy than those who use it as an opportunity to talk, and I wonder if the authors could reflect of their results with regard to these findings in the discussion (there is a result described as patients appreciating the opportunity to learn so the implication is that if they see this as a benefit that clinicians should maximise this; place emphasis on therapy being the opportunity to learn because it has been previously shown that this results in better outcomes than simply using therapy to talk). It is a little unclear how this finding relates to the rationale/objectives and findings of the current study.

Authors' response: We agree that these are important considerations and have addressed this comment by adding statements acknowledging the potential implications of this finding as suggested, both in the 'implications' (p19) and 'unanswered questions and future research' (p20) sections.

Reviewer comment: The authors refer to 'the challenges of therapy' a lot, but it isn't clear what they are referring to from the results – understanding challenges wasn't a key objective so the finding with regard to the objective they relate to (is this acceptability and is it related to the theme of barriers?). I think generally the point is to make sure that the key findings relate to the key objectives so you orient the reader and the reader can be sure that you have addressed the objectives.

Authors' response: The references to 'challenges of therapy' were indeed referring to potential barriers discussed in relation to the research question on acceptability. We have amended the phrasing throughout to make this clearer (p2 line 21, p4 line 19, p16 line 6, p18 line 26).

Reviewer comment: The implications section of the discussion needs to more directly talk about the implications for clinicians – I think this is where the paper has a lot to offer; the implications need to be the focus rather than interpreting the findings from this and other research e.g. what does it mean in terms of implications that patients expect themselves to remain vulnerable and continue to implement techniques learnt during therapy? I think it is important to spell this out for the reader.

Authors' response: We have amended the implications section so that every summary point is directly applied to implications for clinicians (p19). We have also focused more on the implications for clinicians in the unanswered questions and future research section (p20).

Reviewer comment: I think the phrase with regard to the cognitive techniques “was largely discussed by participants who received CBT” (it is stated in the discussion and abstract) needs to be revised; it is expected that only these participants would refer to cognitive techniques, perhaps add this by way of revision.

Authors' response: We agree that this difference is to be expected and have amended the phrasing in the abstract and discussion accordingly (p2 line 14, p16 line 27, p17 line 1).

Reviewer comment: In terms of the limitations with regard to interviewing as soon as possible; I wonder if the other point is that you had difficulty getting those who had completed less than 8 sessions because they dropped out and didn't want to be involved and you could speculate about why this was e.g. it wasn't helpful?

Authors' response: We agree that this was a limitation and have added a comment regarding this as suggested (p18).

Reviewer comment: The authors cite five previous studies that have explored patients' experience of CBT for depression. There are also a fair number of studies that have looked at patients' experiences of CBT for anxiety disorders and other conditions. In order to give the reader a fairer impression of the state of the field, it would have been good if these studies were briefly mentioned. The conclusions are in general similar to the ones that pertain to the studies of CBT for depression.

As far as I am aware, there are no quantitative studies of patients' experience behavioural activation. This is a clear selling point for the present study.

Authors' response: We have added some references to qualitative studies of CBT for other conditions such as anxiety and psychosis, as suggested (p4).

Reviewer comment: It seems that patients report broadly similar experiences as they progress through BA and CBT. However, there are two areas where differences emerge (see pages 9 and 10). The way these differences are described betray the principal investigator's well-known preference for behavioural activation. I would suggest that the authors consider somewhat different phrasing so that a more impartial account is presented. In qualitative methodology one does not usually give percentage figures for the prevalence of particular themes. Instead, the convention is to use just a few, rather vague, adjectives to denote frequency. For example, "all", "many/most", "some/few" and "none". In general the authors stick to this convention. However, in the two places where a somewhat negative theme emerges for behavioural activation that is not present in CBT, the authors use a different language that would appear to be designed to minimise this observation. For example, "a small number" and "a small minority". This terminology is not used elsewhere in the Results section. I suggest the authors revert to using the more general terms that they use for other results.

Alternatively, if they feel strongly that they want to quantify these particular observations, they should give the percentage figures. That would not be the normal convention for qualitative work and would seem special pleading. However, if the authors are particularly keen for readers to know the absolute rates that would be a reasonable alternative way of progressing.

Authors' response: Whilst we take note of the comment relating to the tone of our narrative on this point, we reject the reviewer's implication of bias in the phrase "betray the principal investigator's well-known preference for behavioural activation". The reviewer himself has a well-known preference for CBT but we trust his review is balanced and impartial and from this position we have reflected on his observation. Our investigator team was a mix of clinicians, methodologists and patients with a balance between those that have followed a behavioural and a cognitive path (or neither) in their careers. We ensured that multiple clinical and methodological checks and balances were in place to ensure that neither treatment was given a preferential hearing in our analysis of all our data. The COBRA trial was conducted to the highest GCP and ethical standards, was undertaken by a core team of experts in BOTH BA and CBT, including one of the foremost international experts in CBT from the USA, and was overseen by independent Trial Steering and Data Management committees. Extreme care was taken to ensure that both clinical protocols, including CBT, were designed, delivered, supervised and monitored to the very highest standards by acknowledged clinical experts. Our per-protocol analyses maximised the likelihood of strong clinical performance and outcomes from the gold standard CBT treatment. Methodologically and clinically, therefore, the COBRA trial was conducted to ensure neither treatment was delivered in a way that would have biased the trial against CBT.

We have aimed to report a careful analysis and description of the results of this qualitative sub-study,

undertaken as it was with the same rigour. As the reviewer rightly notes, consistent with a qualitative approach, we did not provide numbers of people with specific responses. The data he refers to is indeed a minority view from a few respondents (2/18, which some may consider to be less than “some” or “few”) who made comments that were different in terms of content from the other respondents. In our original text we wanted to (a) acknowledge those comments, and (b) acknowledge that they were indeed from a very small proportion of the sample, so that readers can make up their minds appropriately in the interest of scientific integrity. Despite our unwillingness to quantify qualitative data, this point is so important that we have now agreed with the reviewer’s suggestion and cited these informant numbers, and have amended the results section to provide numbers of participants expressing this view (p10 line 9, p10 line 14, p11 line 18), to prevent any suggestions that we are underplaying the potential significance of any of our data. We agree that this is unusual, but our “special pleading” is that we absolutely do not wish the clinical community to suspect erroneously that we are being less than objective in reporting our data. Our multidisciplinary and multi-philosophical team regard these findings, their interpretation and our reporting as a true and accurate description of our data.

Reviewer comment: One of the questions in the qualitative interview enquires about behaviour change and another enquires about cognitive change. The Abstract and the Results claim that BA and CBT are both perceived as promoting behaviour change but only CBT promotes cognitive change. This observation seems to rely on a particularly restrictive definition of cognitive change, which is not made explicit in the manuscript but is fairly common among researchers who have a strong allegiance to behavioural activation. The issue is how one categorizes rumination. The authors categorize it as a behaviour, which is how it is usually conceptualised within behavioural analysis. However, most cognitive psychologists and cognitive therapists would classify it as a cognitive process. It would be helpful if this well-known difference in conceptualisation were acknowledged. Authors’ response: We have used this helpful point from the reviewer to clarify our abstract and results sections and indicate that rumination can be considered both a habitual behaviour and involves repetitive negative thinking – indeed both abstract and results make clear that both BA and CBT produce cognitive and behavioural change. We have added a reference to the alternative view of rumination as a cognitive process in the strengths and weaknesses section (p18).

Reviewer comment: The authors mention (page 3) their well-known belief that behavioural activation is easier than CBT to teach to relatively inexperienced and low paid mental health workers. This seems a rather pessimistic view of the intellectual abilities of a very important workforce and, as far as I’m aware, it has never been tested. Given this point, the statement should either be removed or qualified by indicating that it is a belief of the authors that has not been tested. Authors’ response: By suggesting that BA may be easier to teach and learn we do not mean to imply inferior intellectual capacities of either mental health workers or patients. We do not accept that the hypothesis that BA is easier to teach than CBT is our “well-known belief”. US clinical scientists in fact advanced this in the literature some 15 years ago and our trial was funded to investigate the hypothesis that BA could be a viable and parsimonious alternative to CBT. We have now cited the first suggestion of parsimony in our manuscript (p3 line 27, p4 lines 1-2).

Reviewer comment: The patients who formed the sample in this study are recruited from the COBRA trial, which has a confounded design. The trial seems to have been designed to demonstrate that behavioural activation is more cost-effective than CBT. As the two treatments have frequently been shown to achieve similar clinical outcomes, the design more or less guaranteed the cost-effectiveness result that the investigators were looking for by using high paid clinicians to deliver CBT and lower paid clinicians to deliver behavioural activation. A scientific test of the cost effectiveness issue, rather than a demonstration of a particular point of view, would have required an additional cell in which low paid workers delivered CBT. For the current paper, the problem with the confounded design is that it makes it difficult to interpret the occasional differences in findings between BA and CBT. Are the few

negative comments relating to BA but not CBT a true reflection of the experience of BA or a function of the therapy being delivered by a relatively less experienced workforce? The authors seem to hold the view that therapist experience and training is the key issue. That might be correct. However, the contrary view should also be acknowledged as a possibility, especially as there are no other qualitative analyses of behavioural activation.

Authors' response: The reviewer suggests that, "the two treatments have frequently been shown to achieve similar clinical outcomes" but as he well knows, pre-COBRA published trials were of such poor quality and size that NICE could not recommend BA as a treatment for depression. Therefore, it is not true to say that "the design more or less guaranteed the cost-effectiveness result that the investigators were looking for". Clinical non-inferiority of BA to CBT was not already known given the lack of large and high quality trials. The primary purpose of the COBRA trial, therefore, was pragmatic and designed to test whether BA delivered by a low intensity workforce was as efficacious as CBT delivered by the high intensity workforce who are currently contracted to deliver CBT, in response to the hypothesis that BA was not inferior to CBT and can be delivered by less experienced staff. As such, the design was not confounded for this primary question. Indeed, the reviewer's opinion is at odds with all the other reviewers (NIHR peer reviewers, HTA Board, Lancet reviewers and independent experts) who contributed to the design, funding and conduct of our trial. Other trial designs (e.g. including an arm of CBT provided by a low intensity workforce) would answer different questions. We would welcome the results of any such trials, as they would provide further insight into whether BA was not sufficiently complex for some patients, or whether it was the inexperience of the therapists that made the difference.

In terms of the reviewer's comments on cost-effectiveness, we cannot agree. His suggestion that the results of the trial were pre-ordained fails not only to recognise the uncertainty in clinical outcomes referred to above, but also the impact of uncertainty in the multiple variables affecting a health economic analysis. The salary of workers is merely one component of our comprehensive cost-effectiveness analysis. Clinical outcome, number and duration of therapy sessions, and use of other health and social services resources could all have varied in positive or negative ways with reference to BA. It was perfectly possible that therapist costs could have been a minor resource determinant of our analysis compared to these other factors. We entirely reject the suggestion that the trial design has "more or less guaranteed the cost-effectiveness result that the investigators were looking for". Finally, as previously noted, we do accept the reviewer's suggestion that we distinguish between the minority (2/18) respondents who believed BA to be of insufficient complexity to meet their needs and those that referred to the inexperience (also 2/18) of therapists, and have added a statement regarding this in the discussion (p19).

Reviewer comment: The authors state in the Discussion that the perceived focus on behavioural and cognitive change in the treatments led to the observed symptom reductions. This statement goes beyond the data in the study. In order to demonstrate that cognitive or behavioural change mediates symptom change, a different quantitative methodology is required, with multiple measurement of the mediators and statistical modelling that relates the mediators to symptom outcome with suitable temporal lags. There are a fair number of studies that have done exactly this in CBT for depression and there may now also be some such studies for behavioural activation. If the authors want to link behaviour or cognitive change to symptom change, they should refer to such studies. Alternatively, if their statement is simply about patient's perceptions of the effectiveness of behavioural or cognitive change, they should make this clear. Further evidence for these perceived relationships should also be included in the results.

Authors' response: We agree that the statement in the discussion went beyond the data and have amended this to clarify that this conclusion is about patients' perceptions of how behavioural and cognitive change affected symptoms (p16 lines 6-8). We did conduct a quantitative mediation analysis as part of the trial and this will be reported elsewhere, but is beyond the scope of the current paper. We have added a statement regarding this in the discussion (p20). We believe, however, that there is evidence for these patient-perceived relationships in the results. For example, the quotes on p11 and

12 from participants 24, 23 and 12 all provide specific examples of behavioural or cognitive change leading, in their opinion, to them feeling better.

VERSION 2 – REVIEW

REVIEWER	David M Clark Chair of Experimental Psychology Department of Experimental Psychology University of Oxford UK
REVIEW RETURNED	10-Dec-2016

GENERAL COMMENTS	The authors are to be congratulated on a series of revisions that have further strengthened an already strong article. In my original review I suggested six revisions. Four have been implemented in a straightforward manner. In particular:  1) Expanding the Introduction to cover qualitative studies of CBT for other clinical conditions. 2) Consideration of the fact that it is difficult to work out whether the few negative comments about BA relate to the relative inexperience of the therapists or are a characteristic of the treatment itself (revised Discussion). 3) Clarification that rumination can be considered both a habitual behaviour (in the BA framework) and a cognitive process (in the CBT framework). 4) Agreement that the Discussion went beyond the data when talking about the mechanism of change in either treatment. The revised Discussion is entirely appropriate. There are two further revisions I suggested in the review that I would suggest still require some attention. First, it was suggested that the small number of negative responses from participants who received BA should be described within the broad adjectival framework that is used in the rest of this qualitative paper (usual practice) or exact percentage figures should be given (less usual practice but also perfectly fine). The authors have chosen to go with specifying the figures. However, they have not given percentages but have instead given absolute numbers without the denominator in the relevant part of the text. This is perhaps a little misleading as the total sample size is also small. I suggest they either provide the information that they've provided in the cover letter that accompanied their resubmission (i.e. 2/18) or the percentage figure (i.e. 11%). Reading the cover letter I realise that it is also not clear whether the two people who expressed views about the inexperience of their therapists and the two people who believed BA to be insufficiently complex were the same or different individuals. It would help if the authors could clarify this in the text. Second, the original COBRA trial report states that it would be easier to train a BA workforce than a CBT workforce (page 879). In the original submission of the present article the statement was repeated with the authors writing that BA "is proposed to be easier than CBT for patients and practitioners to understand". As far as I'm aware, the first person to make this claim was Neil Jacobson in a series of conference talks in the late 90s (which I attended) and in a (cited) article in 2001. In my original review I pointed out that this
--

claim has not been tested and was concerned that repetition of the claim without any reference to the fact that it is an untested belief is problematic. As someone who trained in both BA and CBT I personally think they are similarly easy to understand and to teach to other people, but I fully appreciate that this is also just a belief! My requested revision was for the authors to either delete the relevant sentence or to qualify it by explaining that the proposal has not yet been tested. The authors have declined to follow either option and instead have beefed up the claim by implying that it has been around for a long time (which it has) and that this justifies the widespread dissemination of BA. (The revised sentence reads: "Furthermore, for fifteen years it has been proposed that BA should be easier than CBT for patients and practitioners to understand and implement, making it an excellent potential candidate for further investigation and widespread dissemination".) I assume the authors didn't really mean to imply that because Neil (who was an old friend) thought he could teach BA more easily, that in itself justifies its widespread dissemination. Could I suggest that the authors follow my original suggestion and mention that the proposal that BA is easier to teach than CBT has still to be tested?

Aside to the authors that requires not further revisions: In my original review I pointed out that the COBRA trial involves a confounded design in experimental terms. This means that it manipulates two potentially important variables simultaneously and in a correlated manner, which inevitably restricts the generality of the conclusions that can be drawn from the findings. In view of the way psychotherapy research is going at the moment, one can easily envision a similarly confounded design concluding that CBT is more cost effective than BA when the outcomes of the two were equivalent but the investigators chose for CBT an even cheaper delivery system than the one the COBRA trial chose for BA. Such a delivery system may be just around the corner given the recent developments (such as Sleepio) in the use of machine learning algorithms to enhance retention and outcomes of internet delivered therapies. I would not think it would be right to conclude the CBT is more cost effective than BA if the design of a trial only compared BA given in face-to-face therapy sessions with junior mental health workers and CBT delivered by a much cheaper internet delivery system enhanced by machine learning. I would want to see a further trial arm in which BA was also delivered by the same cheaper system. Would the authors disagree?

In response to my comment about a confounded design, the authors point out that no other reviewer has mentioned this point to them and then give a detailed account of the strengths of their study. I am fascinated to hear that nobody else has pointed out the confound. To an experimentalist it seems obvious. However, I do want the authors to know that I fully agree with their statements about the general quality of their trial and its importance for a deserved NICE recommendation in favour BA. The COBRA trial is exemplary in almost every respect. My only quibble, which I notice several other readers have also picked up on, concerns the therapist competency ratings. Ideally in a comparative outcome study one would want to have ratings of the competence with which each treatment was delivered and show that this was high and did not differ between the two treatments. Unfortunately, there is no obvious walk across from the competence ratings for BA to those for CBT as very different scales were used for the two treatments. The mean competence rating for CBT on the CTRS (37.9, SD 10.9) is somewhat surprising.

	The minimum pass mark on this scale for IAPT training courses is 36, other CBT training courses adopt a similar or higher pass mark. Assuming a roughly normal distribution, it would appear that around 40% of all of the CBT sessions that were sampled fall below the minimum pass mark. This may have also been true in BA but one can't work that out.
--	--

VERSION 2 – AUTHOR RESPONSE

Reviewer comment: First, it was suggested that the small number of negative responses from participants who received BA should be described within the broad adjectival framework that is used in the rest of this qualitative paper (usual practice) or exact percentage figures should be given (less usual practice but also perfectly fine). The authors have chosen to go with specifying the figures. However, they have not given percentages but have instead given absolute numbers without the denominator in the relevant part of the text. This is perhaps a little misleading as the total sample size is also small. I suggest they either provide the information that they've provided in the cover letter that accompanied their resubmission (i.e. 2/18) or the percentage figure (i.e. 11%). Reading the cover letter I realise that it is also not clear whether the two people who expressed views about the inexperience of their therapists and the two people who believed BA to be insufficiently complex were the same or different individuals. It would help if the authors could clarify this in the text.

Authors' response: We have amended the text so that it now states 2/18, as suggested. We have also confirmed in the text that one of the participants who made these two critiques of BA was the same person; the other two were different individuals.

Reviewer comment: Second, the original COBRA trial report states that it would be easier to train a BA workforce than a CBT workforce (page 879). In the original submission of the present article the statement was repeated with the authors writing that BA "is proposed to be easier than CBT for patients and practitioners to understand". As far as I'm aware, the first person to make this claim was Neil Jacobson in a series of conference talks in the late 90s (which I attended) and in a (cited) article in 2001. In my original review I pointed out that this claim has not been tested and was concerned that repetition of the claim without any reference to the fact that it is an untested belief is problematic. As someone who trained in both BA and CBT I personally think they are similarly easy to understand and to teach to other people, but I fully appreciate that this is also just a belief! My requested revision was for the authors to either delete the relevant sentence or to qualify it by explaining that the proposal has not yet been tested. The authors have declined to follow either option and instead have beefed up the claim by implying that it has been around for a long time (which it has) and that this justifies the widespread dissemination of BA. (The revised sentence reads: "Furthermore, for fifteen years it has been proposed that BA should be easier than CBT for patients and practitioners to understand and implement, making it an excellent potential candidate for further investigation and widespread dissemination".) I assume the authors didn't really mean to imply that because Neil (who was an old friend) thought he could teach BA more easily, that in itself justifies its widespread dissemination. Could I suggest that the authors follow my original suggestion and mention that the proposal that BA is easier to teach than CBT has still to be tested?

Authors' response: We agree with the reviewer's point. We have made it clear in the text (p3-4) that this is an idea that remains to be tested and have amended the text thus: Furthermore, in 2001 it was proposed that BA should be easier than CBT for patients and practitioners to understand and implement.(8) Although this idea remains to be directly tested, BA is an excellent potential candidate for further investigation and possible dissemination if it is shown to be equivalently effective compared with CBT.

Reviewer comment: Aside to the authors that requires not further revisions: In my original review I

pointed out that the COBRA trial involves a confounded design in experimental terms. This means that it manipulates two potentially important variables simultaneously and in a correlated manner, which inevitably restricts the generality of the conclusions that can be drawn from the findings. In view of the way psychotherapy research is going at the moment, one can easily envision a similarly confounded design concluding that CBT is more cost effective than BA when the outcomes of the two were equivalent but the investigators chose for CBT an even cheaper delivery system than the one the COBRA trial chose for BA. Such a delivery system may be just around the corner given the recent developments (such as Sleepio) in the use of machine learning algorithms to enhance retention and outcomes of internet delivered therapies. I would not think it would be right to conclude the CBT is more cost effective than BA if the design of a trial only compared BA given in face-to-face therapy sessions with junior mental health workers and CBT delivered by a much cheaper internet delivery system enhanced by machine learning. I would want to see a further trial arm in which BA was also delivered by the same cheaper system. Would the authors disagree?

In response to my comment about a confounded design, the authors point out that no other reviewer has mentioned this point to them and then give a detailed account of the strengths of their study. I am fascinated to hear that nobody else has pointed out the confound. To an experimentalist it seems obvious. However, I do want the authors to know that I fully agree with their statements about the general quality of their trial and its importance for a deserved NICE recommendation in favour BA. The COBRA trial is exemplary in almost every respect. My only quibble, which I notice several other readers have also picked up on, concerns the therapist competency ratings. Ideally in a comparative outcome study one would want to have ratings of the competence with which each treatment was delivered and show that this was high and did not differ between the two treatments. Unfortunately, there is no obvious walk across from the competence ratings for BA to those for CBT as very different scales were used for the two treatments. The mean competence rating for CBT on the CTRS (37.9, SD 10.9) is somewhat surprising. The minimum pass mark on this scale for IAPT training courses is 36, other CBT training courses adopt a similar or higher pass mark. Assuming a roughly normal distribution, it would appear that around 40% of all of the CBT sessions that were sampled fall below the minimum pass mark. This may have also been true in BA but one can't work that out.

Authors' response: We thank the reviewer for his careful comments which merit some debate, although we have not included any changes to our manuscript since this was not requested regarding these points.

1) Our trial was a pragmatic design. Our non-inferiority design was chosen to test BA delivered more cheaply compared with CBT. To justify a non-inferiority approach, and in response to a published critique of it (Vieta and Sanchez-Moreno, Lancet 2017), in order to justify selection of this design we recently proposed (Richards et al, Lancet 2017) four a priori questions: (1. Is the control treatment effective in meta-analyses? 2. Is it routinely available to patients? 3. Does the untreated condition cause substantial morbidity, mortality, and economic burden? 4. Is the post-randomisation follow-up period substantial?) and one post hoc question (5. Do the trial's control group outcomes compare favourably to effectiveness demonstrated in meta-analyses?). Our COBRA trial met all five criteria. Our choice of CBT delivered in the manner it was, therefore, was driven by our need to compare BA against the gold standard CBT delivery system, which in the UK at least is delivered as we have described it in our trial. Had we chosen a design whereby CBT was also delivered by mental health workers with less clinical training, our trial would not have met the a priori conditions for a non-inferiority trial above and would have been less pragmatically generalisable to the current clinical environment. We might also have been subject to the well-rehearsed criticism of non-inferiority trials (Vieta and Sanchez-Moreno, Lancet 2017), that we were merely comparing two inferior treatments with each other, resulting in a colossal amount of research waste in terms of resources, time and effort. Keeping the CBT arm to the highest standards as possible, was in fact the stiffest test we could have given to BA, with the further potential jeopardy of lesser qualified BA delivery tipping the balance of potential outcomes much more in favour of CBT. In planning our trial, these considerations outweighed the experimental objections outlined above by the reviewer.

2) In terms of competence ratings, the ratings of our random sample of therapy tapes showed that both CBT therapists and BA MHWs were on average performing above competency thresholds. In terms of CBT, our mean sessional competence ratings (37.9) were very similar to the means reported in the COBALT trial, another significant and large recent UK trial of CBT – 38.8, demonstrating that our therapists were achieving competency levels consistent with other similar pragmatic effectiveness studies. We entirely agree with the reviewer that on this random sample of tapes some therapists were scoring below the threshold of competence for that specific session as assessed by our external raters, although we would stress that all therapists had exceeded the competence threshold at the end of their COBRA-specific protocol training course. Our view on this is as follows: our CBT therapists had received one year post-graduate training in CBT, had passed similar competency tests in order to qualify from these courses, had received an additional specialist training in the COBRA protocol, and were supervised by CBT experts. Most importantly, they were NHS employees engaged in routine treatment for patients with depression in the UK NHS IAPT services, indeed they were working for the NHS alongside their trial duties. Finally, the results from the CBT arm in our trial are consistent with other research findings in the field. Therefore, whilst it may be possible to train CBT therapists to achieve higher competency ratings, we suggest that these levels of competence are those actually seen in routine clinical practice in IAPT services in the UK, results achieved following substantial investment in therapists' clinical training. We also agree with the reviewer that the two scales used are not directly comparable. In reference to this point, whilst the measures of competence for the BA MHWs and the CBT therapists are not directly comparable, we note that the MHWs scored on average further above the competency threshold than the CBT therapists on the relevant competency measure. This provides some, albeit weak and indirect, supporting evidence for the discussion above, that it might be easier to train people to be competent BA workers than CBT therapists, although this is moderated by the difficulties with the unknown psychometric properties of the BA competence measure in particular and as rightly pointed out by the reviewer and now acknowledged in our manuscript, this proposition has not been tested head to head.

VERSION 3 – REVIEW

REVIEWER	David M Clark University of Oxford
REVIEW RETURNED	22-Feb-2017
GENERAL COMMENTS	The authors have made the small changes that were requested, further strengthening an already excellent paper